# Chalcogen Bond Involving Zinc(II)/Cadmium(II) Carbonate and Its Enhancement by Spodium Bond

**DOI:** 10.3390/molecules26216443

**Published:** 2021-10-26

**Authors:** Na Liu, Xiaoying Xie, Qingzhong Li

**Affiliations:** The Laboratory of Theoretical and Computational Chemistry, School of Chemistry and Chemical Engineering, Yantai University, Yantai 264005, China; liuna_jiayou00@sina.com

**Keywords:** chalcogen bonds, spodium bonds, synergistic effect, QTAIM, NBO

## Abstract

Carbonate MCO_3_ (M = Zn, Cd) can act as both Lewis acid and base to engage in a spodium bond with nitrogen-containing bases (HCN, NHCH_2_, and NH_3_) and a chalcogen bond with SeHX (X = F, Cl, OH, OCH_3_, NH_2_, and NHCH_3_), respectively. There is also a weak hydrogen bond in the chalcogen-bonded dyads. Both chalcogen and hydrogen bonds become stronger in the order of F > Cl > OH > OCH_3_ > NH_2_ > NHCH_3_. The chalcogen-bonded dyads are stabilized by a combination of electrostatic and charge transfer interactions. The interaction energy of chalcogen-bonded dyad is less than −10 kcal/mol at most cases. Furthermore, the chalcogen bond can be strengthened through coexistence with a spodium bond in N-base-MCO_3_-SeHX. The enhancement of chalcogen bond is primarily attributed to the charge transfer interaction. Additionally, the spodium bond is also enhanced by the chalcogen bond although the corresponding enhancing effect is small.

## 1. Introduction

Zinc and cadmium are two of the most widely used non-ferrous metals because of their unique physical and chemical properties [1]. They have been explored mainly in natural minerals in the form of carbonates. Zincite ZnCO_3_ and calcite CdCO_3_ are the most common carbonate rock-forming minerals. MCO_3_ (M = Zn, Cd) possesses a planar triangular configuration with *p*-conjugated molecular orbitals, which is similar to (BO_3_)^3−^, (B_3_O_6_)^3−^, and (B_3_O_7_) [2,3,4]. MCO_3_ has been accepted as a promising material for optical devices in the ultraviolet region due to its fascinating crystal structure, high birefringence, nonlinear optical properties, and transparency [5,6,7]. Owing to their special electronic energy band structures, these carbonates also have drawn extensive attention in the fields of chemical sensing, luminescence, lithium ion battery anode materials, catalysis, and magnetic materials [8,9,10]. The spodium bond (SpB) is a non-covalent interaction that occurs between the Group 12 metal atom and a Lewis base [11]. A region with positive molecular electrostatic potentials (MEPs) is found on the metal atomic surface of tetracoordinated molecules CdBr_2_L_2_ and HgCl_2_L_2_ (L = thiourea) [12], thus it can form a SpB with some bases such as CO, CH_3_CN, H_2_CS, and H_2_CO, with interaction energy less than 9 kcal/mol. A SpB was found for bicoordinated molecules [13], which is stronger than the tetracoordinated molecule. Recently, Liu and Li [14] studied the spodium-bonded complexes of MCO_3_ (M = Zn, Cd, and Hg) with three N-bases (HCN, NHCH_2_, and NH_3_) and found that they are more bonded than other types of M(II) molecules since the interaction energy exceeds 30 kcal/mol.

Se compound is not only taken as an electron donor in non-covalent interactions owing to its lone pair electrons but also acts as a Lewis acid in chalcogen bond, which is usually used to describe the non-covalent interaction between the σ-hole/π-hole of the seventh main group atom and Lewis bases [15]. The σ-hole bonding interaction was first utilized to describe halogen bonding [16]. Then such σ-hole bonding was extended to molecules containing group VI atoms [17]. This concept was further reviewed and revisited [18,19]. Organoselenium compounds can engage in a chalcogen bond with different electron donors such as N, O, and halogen through an intermolecular or an intramolecular mode [20]. In most cases, the Se⋯O chalcogen bond is not very strong with interaction energy less than 10 kcal/mol [21]. For the O electron donors, they are often from H_2_O and its derivative as well as carbonyl compounds. However, the chalcogen bond between Se compounds and carbonate MCO_3_ (M = Zn, Cd) has not been reported.

An important property of a non-covalent interaction is cooperativity, which can be divided into positive and negative cooperativity. When multiple non-covalent interactions coexist in the same system, their strengths will be enhanced or weakened, thus resulting in a cooperative effect. The cooperativity is of great importance in chemical reactions, molecular recognition, and biochemical regulation [22,23]. Some studies showed that the chalcogen bond displays cooperativity with itself [24,25,26], halogen bond [27], hydrogen bond [28], alkaline-earth bond [29], and cation-π [30] interactions. However, few studies have been performed for the cooperativity of chalcogen bond with spodium bond.

In this study, we explore the chalcogen bond formed by carbonate MCO_3_ (M = Zn, Cd) with SeHX (X = F, Cl, OH, OCH_3_, NH_2_, and NHCH_3_) to unveil the substituent effect on the strength of chalcogen bond and its origin. Then adding a spodium bond to the M atom of MCO_3_ in MCO_3_-SeHX, we study the cooperative effect between the chalcogen and spodium bonds, particularly the enhancing effect of spodium bond on the chalcogen bond, and its mechanism is explored by means of MEP and charge transfer. These dyads and triads have been analyzed in the geometries, energetic, AIM, charge transfer, and orbital interaction.

## 2. Theoretical Methods

All calculations were carried out using the Gaussian 09 program (Gaussian, Inc., Wallingford, CT, USA) [31]. The structures of all monomers and complexes were optimized at the MP2/aug-cc-pVTZ level. Basis set aug-cc-pVTZ-PP including pseudopotential was applied for Zn and Cd atoms to account for relativistic effects [32]. Then, harmonic frequency analysis was performed at the same level to ensure that all structures are minima on the potential energy surfaces. The interaction energy was defined as the energy difference between the complex and the monomers, in which their geometries were taken from the complex. The interaction energies were corrected for the basis set superposition error (BSSE) by the counterpoise procedure of Boys and Bernardi [33].

The wave function analysis-surface analysis suite (WFA-SAS) program [34] was used to analyze the molecular electrostatic potential (MEP) of monomers and dyads on the 0.001e Bohr^−3^ isosurface. Topological properties were derived from the theory of atoms in molecules (AIM) using QTAIM software [35]. Natural bond orbital (NBO) analysis was performed at the HF/aug-cc-pVTZ(PP) level by the NBO5.0 program [36], which can provide some information for charge transfer, while orbital interaction was analyzed using the NBO6.0 program [37]. Multiwfn and VMD [38,39] were utilized to map non-covalent interactions (NCI) [40]. To gain further insight into the nature of the investigated intermolecular interactions, energy decomposition analysis (EDA) was conducted using the GAMESS program [41,42].

## 3. Results and Discussion

### 3.1. Chalcogen-Bonded Dyads Involving MCO_3_

Figure 1 shows a diagram of 12 chalcogen-boned dyads of MCO_3_-SeHX (X = F, Cl, OH, OCH_3_, NH_2_, and NHCH_3_, M = Zn^2+^ and Cd^2+^), respectively represented from ChB-1 to ChB-12. In each dyad, there are both a ChB and a HB, where the ChB is stronger than the HB. Both the ChB and HB jointly maintain the stability of MCO_3_-SeHX. Table 1 summarizes their geometric parameters including the angles of X-Se⋯O (α_1_) and Se-H⋯O (α_2_), distances of Se⋯O (R_ChB_) and H⋯O (R_HB_), and changes of C=O (r_1_) and Se-X (r_2_) bond lengths. Both α_1_ and α_2_ angles reflect the direction of ChB and HB, respectively. Both angles vary in ranges of 160–170° and 120–140°, respectively, and the former is bigger than the latter. Accordingly, the ChB is more easily formed and the HB is weak. Moreover, both angles display a slight dependence on the M atom of MCO_3_. Interestingly, both angles vary inversely (Appendix A). That is, the enhancement of ChB is at the sacrifice of the HB weakening, displaying negative cooperativity. Although the ChB is stronger than the HB, R_ChB_ is longer than R_HB_ due to the bigger atomic radius of Se. Both R_ChB_ and R_HB_ become shorter in the CdCO_3_ complex than in the ZnCO_3_ analogue since the O atom of CdCO_3_ has the greater negative MEP [14]. When MCO_3_ is fixed, R_ChB_ is shorter in sequence of NHCH_3_ > NH_2_ > OCH_3_ > OH > Cl > F, while R_HB_ is longer irregularly. Both C=O and Se-X bonds are lengthened in the complexes, and the Se-X bond suffers the larger elongation than the C=O bond due to the fact that the C=O bond is double and there is an increase of charge density on the Se-X anti-bonding orbital.

The last column of Table 1 lists the interaction energy of chalcogen-bonded dyad, which ranges from −3.6 kcal/mol in ChB-11 to −12.3 kcal/mol in ChB-2. There are some regular variations for the interaction energy. Firstly, for a fixed MCO_3_, the interaction energy becomes more negative in the sequence of NHCH_3_ < NH_2_ < OCH_3_ < OH < Cl < F, which is consistent with the positive MEPs on the Se and H atoms (Figure 2). Secondly, comparing SeHOCH_3_ complex with SeHOH analogue, or SeHNHCH_3_ complex with SeHNH_2_ analogue, it is found that the methyl group in the chalcogen donor reduces the interaction energy, showing the electron-donating role of the methyl group (confirmed by the increase of positive MEPs on the Se and H atoms). This weakening effect of methyl group is the same as that in the CH∙∙∙O HB [43]. Thirdly, the interaction energy is larger in the CdCO_3_ complex than that in the ZnCO_3_ analogue, evidenced by the more negative MEP on the O atom of CdCO_3_ [14]. The interaction energy between MCO_3_ and SeHX is larger than that with H_2_O [44], thus the O atom of MCO_3_ is a good electron donor in the chalcogen bond.

Table 2 lists some important AIM parameters including electron density (ρ), its Laplacian (∇^2^ρ), and energy density (H) at the Se∙∙∙O BCP in the chalcogen-bonded dyads. ρ is in the range of 0.012–0.035 a.u., and it displays a quadratic relationship with the Se∙∙∙O distance (Figure 3), with a correlation coefficient of 0.996. Thus the electron density at the Se∙∙∙O BCP can be used to estimate the change of ChB strength. The trend of density Laplacian is the same as the ρ. H is greater than zero in most dyads except ChB-1, ChB-2, and ChB-4. The negative H in ChB-1, ChB-2, and ChB-4 means that the ChB is a partially covalent interaction, consistent with the larger interaction energy (>10 kcal/mol). Although the AIM parameters of H∙∙∙O BCP are not analyzed, its coexistence with the ChB is obviously observed in the NCI diagram (Appendix A), where a blue or green area is present between the bonded two atoms.

A charge density moves from MCO_3_ to SeHX, which is in a range of 0.009–0.005 e (Table 3). There are two types of ChB and HB in each complex, but the direction of charge transfer is the same for both bonds, thus a linear relationship is present between the interaction energy and charge transfer (Figure 4), with a correlation coefficient of 0.994. For the ChB, the charge transfer is moved from the O lone pair orbital (Lp_O_) into the Se-X anti-bonding orbital (σ*_Se-X_), i.e., Lp_O_→σ*_Se-X_. This orbital interaction results in the elongation of the Se-X and C=O bonds. Likely, the perturbation energy of Lp_O_→σ*_Se-X_ orbital interaction also displays a linear relationship with the interaction energy (Appendix A). The HB is characterized with an orbital interaction of Lp_O_→σ*_Se-H_, which is much weaker than that in the ChB, thus MCO_3_-SeHX is primarily stabilized by the ChB. CdCO_3_ engages in a stronger HB than ZnCO_3_, confirmed by the larger perturbation energy of Lp_O_→σ*_Se-H_ in the CdCO_3_ complex.

The interaction energy was decomposed into electrostatic (E^es^), exchange (E^ex^), repulsion (E^rep^), polarization (E^pol^) and dispersion energies (E^disp^), collected in Table 4. The total interaction energy obtained with GAMESS program is almost equal to that with Gaussian program. Among the three attractive terms (E^es^, E^pol^, and E^disp^), E^es^ is largest, confirming that electrostatic interaction is dominant. This conclusion is further confirmed by the linear relationship between the total interaction energy and the electrostatic interaction (Appendix A). No linear relationship is present between the total interaction energy and the other attractive terms. Politzer and co-authors explained most σ-hole interactions by means of Coulombic interactions and concluded that charge transfer is an extreme form of polarization [45,46]. However, the interaction energy of chalcogen bond displays a good linear relationship with the charge transfer but not with polarization energy. Both E^pol^ and E^disp^ are comparable since the latter corresponds to 50–90% of the former.

### 3.2. Cooperativity between Spodium and Chalcogen Bonds in Triads

Only three chalcogen-bonded dyads of ChB-1, ChB-2, and ChB-4 have a larger interaction energy exceeding 10 kcal/mol, thus we are interested in how to strengthen the ChB by adding a spodium bond (SpB) and its enhancing mechanism. Figure 5 shows the diagram of the ternary complex, where a M⋯N SpB coexists with a ChB and a HB. The formation of SpB can be understood by the MEP maps of MCO_3_ and three N-bases (Appendix A), where a red region with positive MEPs and a blue one with negative MEPs are found on the M and N atoms, respectively. Since such M∙∙∙N SpB was analyzed in the previous study [14], thus it is not studied here and our aim is to strengthen the chalcogen bond by means of SpB. These triads are marked from T-1 to T-36. The molecular configuration in the triad is similar to that in the dyad. The angles of X-Se⋯O (α_1_) and Se-H⋯O (α_2_) are almost not changed in the triad relative to the dyad in spite of no regular variation (Appendix A).

Both the ChB and HB interactions are enhanced in the ternary complexes, which can be clearly evidenced by the shorter binding distances (Table 5). The shortening of both Se⋯O and H⋯O distances is very prominent and the largest shortening is up to 0.1 Å for each distance. The largest shortening of Se⋯O distance is found in the triads involving SeHCl. Therefore, an introduction of a spodium bond to MCO_3_ leads to a prominent change in the binding distances of the ChB and HB though a slight change takes place for the angles of X-Se⋯O (α_1_) and Se-H⋯O (α_2_). The shortening of Se⋯O distance is larger than that of H⋯O distance in most triads excluding T-16 and from T-25 to T-36, and their largest difference is up to 0.057 Å. On the other hand, the SpB binding distance is also shortened in the triads although its shortening is much smaller than that of ChB and HB. This indicates that all bonds of ChB, HB, and SpB strengthened each other.

The first column in Table 6 is the total interaction energy of triad, ranging from −37.36 kcal/mol in T-36 to −68.28 kcal/mol in T-3. The interaction energy between SeHX and the N base (∆E_far_) was deducted in calculating the interaction energies of SpB and ChB although this value is very small (<0.4 kcal/mol). Both ∆E_SpB_ and ∆E_ChB_ are increased in the triads relative to the corresponding dyads and their increase is almost equal in most triads. However, the increased percentage is larger for ∆E_ChB_ due to its smaller crude value. This shows that the enhancement of ChB and HB is larger than that of SpB, which supports the previous conclusion that the stronger interaction has a larger effect on the weaker one [47]. Similarly, the largest increased percentage of ChB interaction energy is found in the triads involving SeHCl.

The interplay between different types of bonds in the triad can be estimated with cooperative energy (E_coop_), which was calculated with the formulas of E_coop_ = ∆*E*_total,T_ − ∆E_ChB,D_ − ∆E_SpB,D_ − ∆E_far_, in which ∆E_total,T_ is the total interaction energy of a triad, ∆E_ChB,D_ the interaction energy of the optimized chalcogen-bonded dyad, and ∆E_SpB,D_ the interaction energy of the optimized spodium-bonded dyad. This value is positive in all triads, confirming the positive cooperativity. Moreover, E_coop_ accounts for 2–10% of the total interaction energy, and this ratio falls within 6% of HB cooperativity [48].

The enhancement of ChB and SpB can also be confirmed by the larger electron densities at the Se⋯O and M⋯N BCPs of chalcogen and spodium bonds in the ternary systems compared to their binary analogues (Appendix A). Interestingly, the largest increase in the electron density at the Se⋯O BCP is found in the triads involving SeHCl. Both Laplacians and energy densities are also varied in the triads, but no essential change is found in most triads with an exception in the triads involving SeHCl (Appendix A). For the latter, the energy density at the Se⋯O BCP varies from positive in the chalcogen-bonded dyad to negative in the triad.

As presented in Appendix A, both the positive MEP on the M atom and the negative MEP on the O atom of MCO_3_ are increased when it forms a chalcogen bond and a spodium bond, respectively. This means that the M atom of MCO_3_ in the chalcogen-bonded dyad engages in a stronger SpB, while the O atom of MCO_3_ in the spodium-bonded dyad participates in a stronger ChB and HB. Thus, the positive cooperativity can be explained by a “pull-push” model, in which the SeHX molecule draws more electrons and the N-containing base donates more electrons simultaneously in the ternary complexes. The increase of ∆*E*_ChB_ in the triad is attributed to the contribution from the ChB and HB interactions, thus we explore the relationship between the increase of electron density at the Se⋯O BCP (∆ρ_ChB_, Appendix A) and the increase of negative MEP on the O atom of spodium-bonded dyad (ΔV_S,min_, Appendix A). However, no consistent change is found between both terms, and even a reverse change is observed for them. For example, with the increase of ΔV_S,min_ on the O atom from H_3_N-ZnCO_3_ to H_2_CHN-ZnCO_3_ to HCN-ZnCO_3_, ∆ρ_ChB_ is reduced from 0.0079 a.u. in T = 6 to 0.0075 a.u. in T = 3 to 0.0071 a.u. in T-1. Thus the electrostatic interaction is only used to explain qualitatively the enhancement of ChB by the SpB.

Appendix A presents the charge transfer between MCO_3_ and SeHX (CT_ChB_) as well as between MCO_3_ and N-base (CT_SpB_) in the triads. CT_SpB_ exceeds 0.1e in most triads and is much bigger than CT_ChB_. CT_ChB_ is half of CT_SpB_ for the strong ChB, while the former is less than 10% of the latter for the weak ChB. Both CT_ChB_ and CT_SpB_ are increased in the triads relative to the dyads. Moreover, the increase of CT_ChB_ is much larger than that of CT_SpB_. Being consistent with the increase of the chalcogen bonding interaction energy, the increase of charge transfer for the ChB is largest in the triads involving SeHCl. A linear relationship is found between the increase of CT_ChB_ and the increase of chalcogen-bonded interaction energy (Figure 6), thus the enhancement of ChB and HB is attributed to the increase of charge transfer.

## 4. Conclusions

Systematic theoretical calculations were performed to study the ternary complex of N-base-MCO_3_-SeHX (X = F, Cl, OH, OCH_3_, NH_2_, NHCH_3_; M = Zn, Cd; N-base = HCN, NHCH_2_, and NH_3_) and their corresponding binary analogues. Carbonate MCO_3_ can form not only a spodium bond with the N-base but also a chalcogen bond with SeHX. The spodium bond is stronger than the chalcogen bond since the Zn(II) and Cd(II) are metal cations in the neutral carbonate salts. The chalcogen-bonded dyad is more stable in the sequence of F > Cl > OH > OCH_3_ > NH_2_ > NHCH_3_ and CdCO_3_ > ZnCO_3_. The interaction energy of chalcogen-bonded dyad is greatly affected by the substituent X of SeHX, becoming more negative from −3.6 kcal/mol in ZnCO_3_-SeHNHCH_3_ to −12.3 kcal/mol in CdCO_3_-SeHF. Both electrostatic and charge transfer interactions are responsible for the stability of chalcogen-bonded dyad. Both chalcogen and spodium bonds strengthened each other in N-base-MCO_3_-SeHX, and the enhancing effect is larger for the chalcogen bond, which is mainly attributed to the increase of charge transfer.

## Figures and Tables

**Figure 1 molecules-26-06443-f001:**
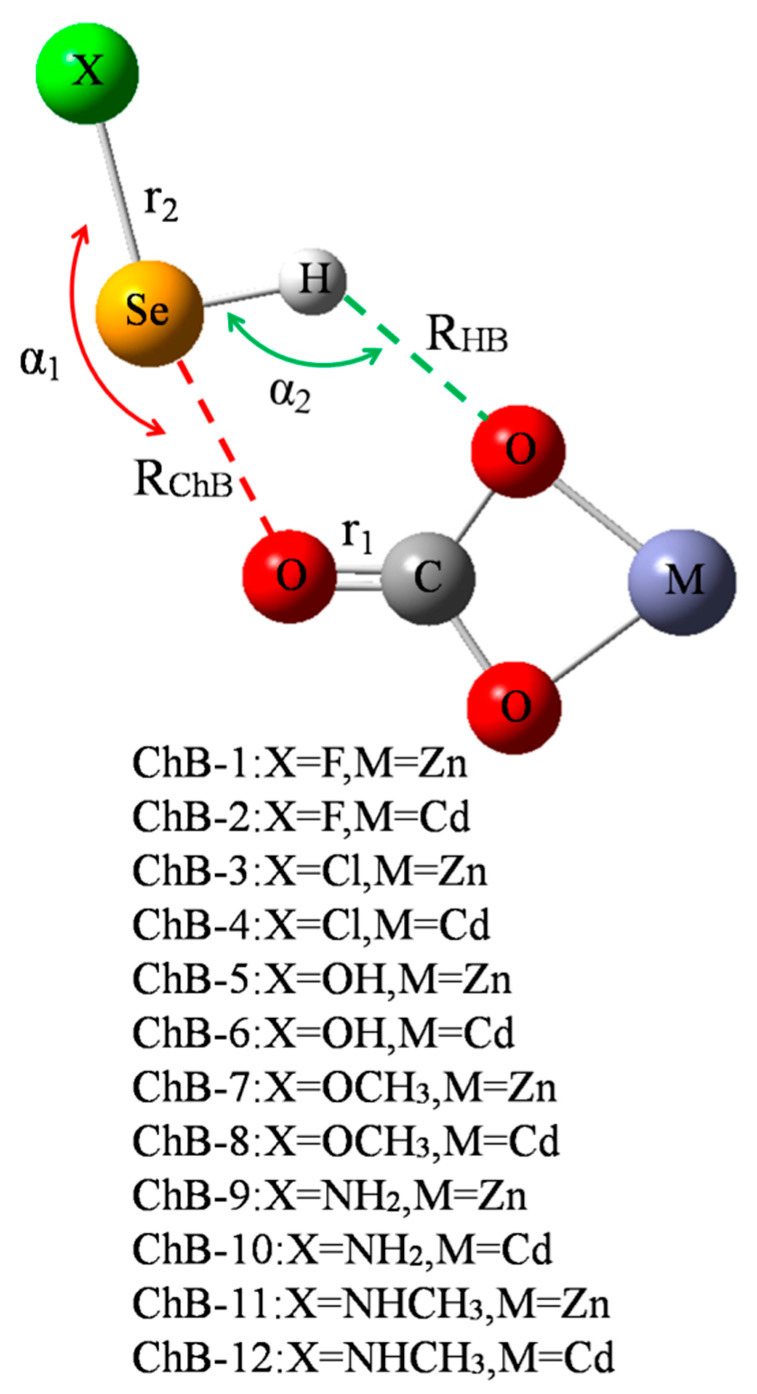
Scheme of chalcogen-bonded dyad.

**Figure 2 molecules-26-06443-f002:**
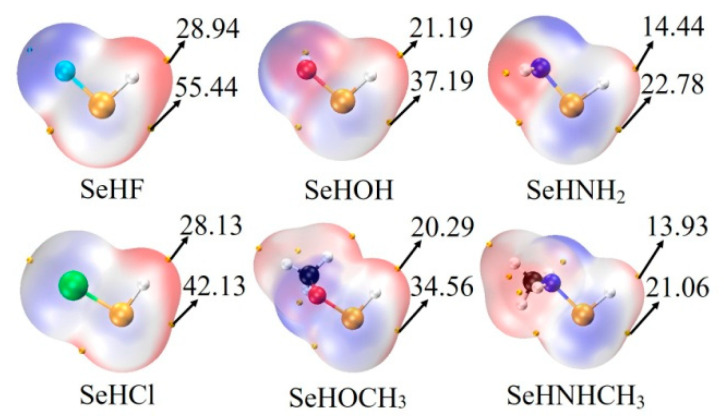
MEP maps on the 0.001 electrons Bohr^−3^ isodensity surface of monomers. Red and blue regions represent positive and negative MEPs, respectively. All are in kcal/mol.

**Figure 3 molecules-26-06443-f003:**
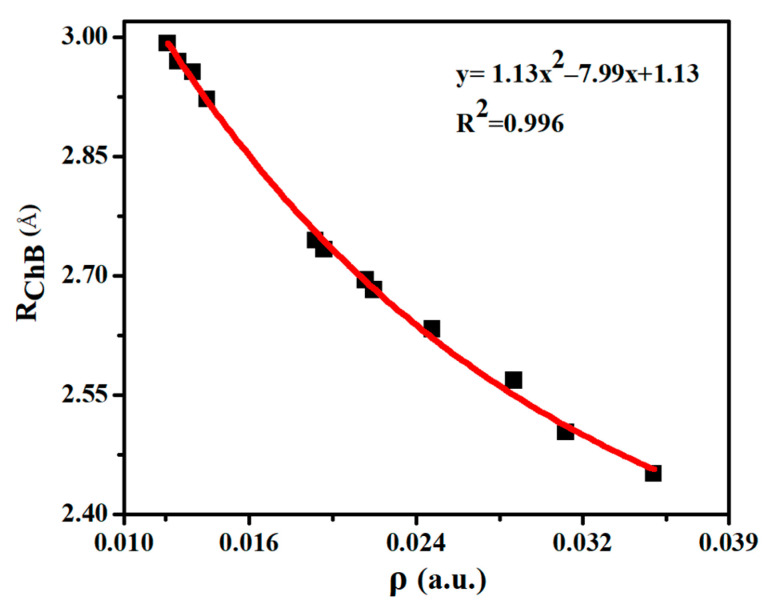
Binding distances (R_ChB_) versus electron density (ρ) at the Se∙∙∙O BCP in the chalcogen-bonded dyads.

**Figure 4 molecules-26-06443-f004:**
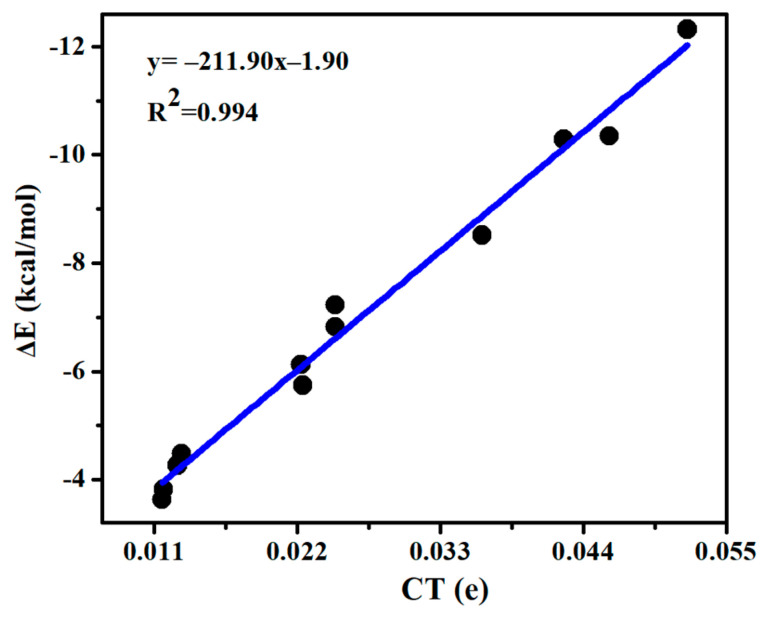
Interaction energy (∆E) versus charge transfer (CT) in the chalcogen-bonded dyads.

**Figure 5 molecules-26-06443-f005:**
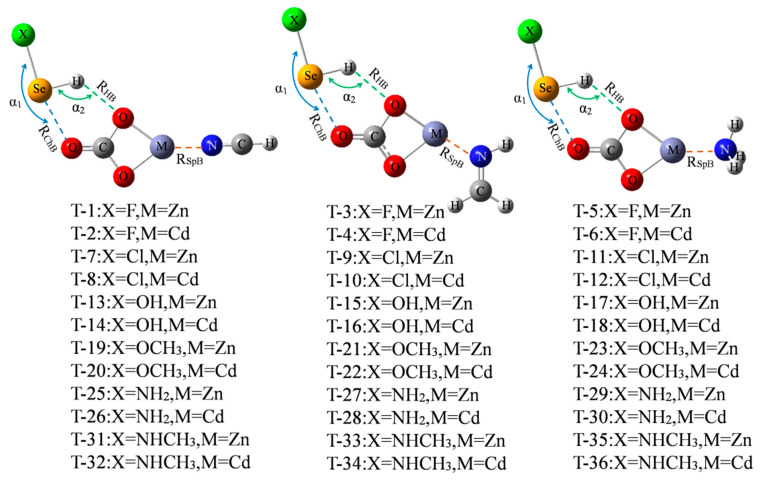
Schemes of XHSe-MCO_3_-N-base ternary complexes.

**Figure 6 molecules-26-06443-f006:**
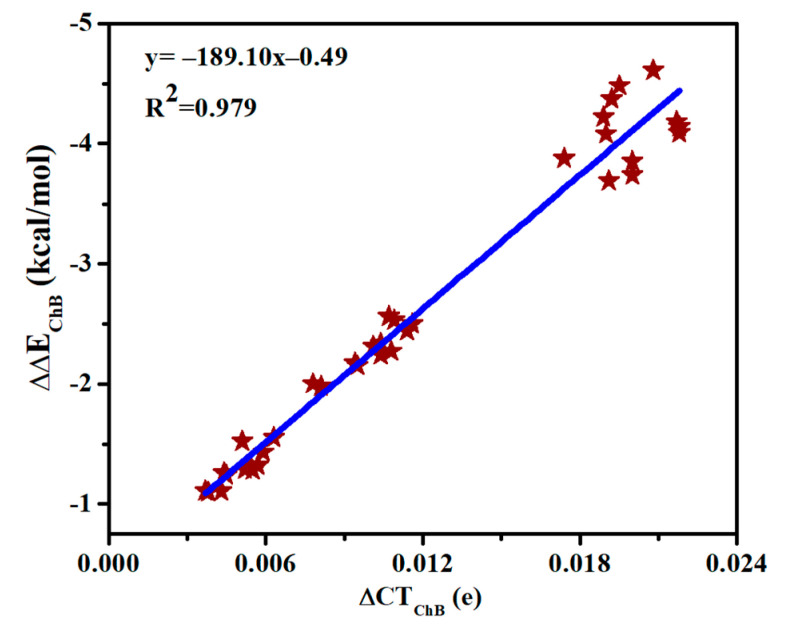
Relationship between increase of both charge transfer (∆CT) and interaction energy (∆∆E) in the chalcogen bonds.

**Table 1 molecules-26-06443-t001:** Angles (α, deg), binding distances (R, Å), change of bond length (Δr, Å), and interaction energy (∆E, kcal/mol) in the chalcogen-bonded dyads.

	α_1_	α_2_	R_ChB_	R_HB_	Δr_1_	Δr_2_	∆E
ZnCO_3_-SeHF(ChB-1)	169.1	125.0	2.503	2.413	0.011	0.032	−10.35
CdCO_3_-SeHF(ChB-2)	169.3	124.5	2.451	2.363	0.012	0.040	−12.32
ZnCO_3_-SeHCl(ChB-3)	166.7	128.8	2.633	2.344	0.009	0.036	−8.52
CdCO_3_-SeHCl(ChB-4)	167.1	128.2	2.568	2.289	0.010	0.045	−10.29
ZnCO_3_-SeHOH(ChB-5)	166.7	129.8	2.733	2.438	0.006	0.021	−6.13
CdCO_3_-SeHOH(ChB-6)	166.9	129.8	2.682	2.380	0.006	0.026	−7.23
ZnCO_3_-SeHOCH_3_(ChB-7)	166.8	130.4	2.744	2.437	0.005	0.018	−5.74
CdCO_3_-SeHOCH_3_(ChB-8)	166.7	130.5	2.694	2.368	0.006	0.023	−6.82
ZnCO_3_-SeHNH_2_(ChB-9)	159.8	138.9	2.969	2.370	0.003	0.015	−3.82
CdCO_3_-SeHNH_2_(ChB-10)	160.0	138.9	2.922	2.313	0.003	0.018	−4.48
ZnCO_3_-SeHNHCH_3_(ChB-11)	159.8	139.6	2.992	2.361	0.002	0.012	−3.62
CdCO_3_-SeHNHCH_3_(ChB-12)	159.2	140.8	2.956	2.286	0.002	0.015	−4.26

**Table 2 molecules-26-06443-t002:** Electron density (ρ), its Laplacian (∇^2^ρ), and energy density (H) at the Se∙∙∙O BCP in the chalcogen-bonded dyads, all in au.

	ρ	∇^2^ρ	H
ChB-1	0.0312	0.1066	−0.0004
ChB-2	0.0354	0.1143	−0.0017
ChB-3	0.0248	0.0867	0.0010
ChB-4	0.0287	0.0959	0.0002
ChB-5	0.0196	0.0734	0.0016
ChB-6	0.0220	0.0803	0.0013
ChB-7	0.0192	0.0718	0.0016
ChB-8	0.0216	0.0784	0.0014
ChB-9	0.0126	0.0486	0.0017
ChB-10	0.0140	0.0532	0.0017
ChB-11	0.0121	0.0466	0.0017
ChB-12	0.0133	0.0504	0.0017

**Table 3 molecules-26-06443-t003:** Charge transfer (CT, e) and second-order perturbation energies (E^(2)^, kcal/mol) in the chalcogen-bonded dyads.

	CT	Type (ChB)	E^(2)^	Type (HB)	E^(2)^
ChB-1	0.0390	LP_O_→σ*_Se-F_	15.12	LP_O_→σ*_Se-H_	0.92
ChB-2	0.0493	LP_O_→σ*_Se-F_	17.25	LP_O_→σ*_Se-H_	1.46
ChB-3	0.0322	LP_O_→σ*_Se-Cl_	10.21	LP_O_→σ*_Se-H_	0.63
ChB-4	0.0404	LP_O_→σ*_Se-Cl_	12.41	LP_O_→σ*_Se-H_	1.08
ChB-5	0.0180	LP_O_→σ*_Se-O_	5.85	LP_O_→σ*_Se-H_	0.25
ChB-6	0.0232	LP_O_→σ*_Se-O_	6.32	LP_O_→σ*_Se-H_	0.42
ChB-7	0.0186	LP_O_→σ*_Se-O_	5.92	LP_O_→σ*_Se-H_	0.26
ChB-8	0.0237	LP_O_→σ*_Se-O_	6.11	LP_O_→σ*_Se-H_	0.40
ChB-9	0.0091	LP_O_→σ*_Se-N_	2.58	LP_O_→σ*_Se-H_	0.13
ChB-10	0.0120	LP_O_→σ*_Se-N_	2.63	LP_O_→σ*_Se-H_	0.14
ChB-11	0.0092	LP_O_→σ*_Se-N_	2.42	LP_O_→σ*_Se-H_	0.14
ChB-12	0.0118	LP_O_→σ*_Se-N_	2.50	LP_O_→σ*_Se-H_	0.14

**Table 4 molecules-26-06443-t004:** Electrostatic (E^es^), exchange (E^ex^), repulsion (E*^r^*^ep^), polarization (E^pol^), and dispersion (E^disp^) energies in the chalcogen-bonded dyads, all in kcal/mol.

	E^es^	E^ex^	E^rep^	E^pol^	E^disp^	E_total_
ChB-1	−35.30	−33.47	73.69	−10.38	−5.22	−10.68
ChB-2	−42.36	−39.88	90.26	−12.01	−8.70	−12.69
ChB-3	−28.83	−31.55	66.06	−9.64	−5.21	−9.17
ChB-4	−35.15	−38.04	80.26	−10.69	−7.00	−10.62
ChB-5	−22.40	−18.74	49.95	−9.00	−6.70	−6.89
ChB-6	−26.82	−23.50	62.09	−12.33	−6.92	−7.48
ChB-7	−20.27	−18.25	47.74	−8.48	−6.92	−6.18
ChB-8	−24.35	−22.38	57.20	−11.29	−6.95	−7.77
ChB-9	−13.07	−19.65	41.35	−7.31	−5.41	−4.09
ChB-10	−15.45	−18.49	43.39	−7.45	−6.63	−4.63
ChB-11	−12.28	−18.52	39.22	−6.90	−5.48	−3.96
ChB-12	−14.32	−17.57	41.47	−7.33	−6.58	−4.33

**Table 5 molecules-26-06443-t005:** Change of binding distance (∆R, Å) in the triads relative to the corresponding dyads.

	∆R_ChB_	∆R_HB_	∆R_SpB_
HCN-ZnCO_3_-SeHF(T-1)	−0.085	−0.070	−0.005
HCN-CdCO_3_-SeHF(T-2)	−0.072	−0.051	−0.028
H_2_CHN-ZnCO_3_-SeHF(T-3)	−0.090	−0.072	−0.005
H_2_CHN-CdCO_3_-SeHF(T-4)	−0.082	−0.062	−0.018
H_3_N-ZnCO_3_-SeHF(T-5)	−0.093	−0.077	−0.006
H_3_N-CdCO_3_-SeHF(T-6)	−0.080	−0.061	−0.030
HCN-ZnCO_3_-SeHCl(T-7)	−0.112	−0.071	−0.005
HCN-CdCO_3_-SeHCl(T-8)	−0.100	−0.054	−0.027
H_2_CHN-ZnCO_3_-SeHCl(T-9)	−0.118	−0.071	−0.005
H_2_CHN-CdCO_3_-SeHCl(T-10)	−0.115	−0.058	−0.017
H_3_N-ZnCO_3_-SeHCl(T-11)	−0.122	−0.080	−0.006
H_3_N-CdCO_3_-SeHCl(T-12)	−0.112	−0.056	−0.026
HCN-ZnCO_3_-SeHOH(T-13)	−0.084	−0.083	−0.003
HCN-CdCO_3_-SeHOH(T-14)	−0.072	−0.062	−0.025
H_2_CHN-ZnCO_3_-SeHOH(T-15)	−0.088	−0.085	−0.003
H_2_CHN-CdCO_3_-SeHOH(T-16)	−0.067	−0.070	−0.024
H_3_N-ZnCO_3_-SeHOH(T-17)	−0.092	−0.090	−0.004
H_3_N-CdCO_3_-SeHOH(T-18)	−0.080	−0.071	−0.027
HCN-ZnCO_3_-SeHOCH_3_(T-19)	−0.086	−0.082	−0.003
HCN-CdCO_3_-SeHOCH_3_(T-20)	−0.077	−0.053	−0.025
H_2_CHN-ZnCO_3_-SeHOCH_3_(T-21)	−0.090	−0.084	−0.003
H_2_CHN-CdCO_3_-SeHOCH_3_(T-22)	−0.071	−0.060	−0.024
H_3_N-ZnCO_3_-SeHOCH_3_(T-23)	−0.095	−0.088	−0.003
H_3_N-CdCO_3_-SeHOCH_3_(T-24)	−0.087	−0.065	−0.024
HCN-ZnCO_3_-SeHNH_3_(T-25)	−0.063	−0.085	−0.002
HCN-CdCO_3_-SeHNH_3_(T-26)	−0.046	−0.079	−0.023
H_2_CHN-ZnCO_3_-SeHNH_3_(T-27)	−0.069	−0.085	−0.002
H_2_CHN-CdCO_3_-SeHNH_3_(T-28)	−0.047	−0.075	−0.022
H_3_N-ZnCO_3_-SeHNH_3_(T-29)	−0.066	−0.100	−0.002
H_3_N-CdCO_3_-SeHNH_3_(T-30)	−0.049	−0.092	−0.024
HCN-ZnCO_3_-SeHNHCH_3_(T-31)	−0.061	−0.090	−0.002
HCN-CdCO_3_-SeHNHCH_3_(T-32)	−0.052	−0.073	−0.023
H_2_CHN-ZnCO_3_-SeHNHCH_3_(T-33)	−0.067	−0.088	−0.002
H_2_CHN-CdCO_3_-SeHNHCH_3_(T-34)	−0.049	−0.066	−0.022
H_3_N-ZnCO_3_-SeHNHCH_3_(T-35)	−0.063	−0.106	−0.002
H_3_N-CdCO_3_-SeHNHCH_3_(T-36)	−0.035	−0.092	−0.027

**Table 6 molecules-26-06443-t006:** Total interaction energy (∆E_total_), interaction energy between molecular pairs (∆E), and cooperative energy (E_coop_) in the ternary complexes, all in kcal/mol.

	∆E_total_	∆E_SpB_	∆E_ChB_	∆E_far_	∆∆E_SpB_	∆∆E_ChB_	E_coop_
T-1	−53.24	−42.42	−14.43	−0.42	−3.97	−4.08	−4.02
T-2	−48.20	−35.42	−16.20	−0.36	−3.58	−3.88	−3.68
T-3	−68.28	−57.39	−14.57	−0.37	−3.94	−4.22	−4.11
T-4	−60.32	−47.48	−16.69	−0.34	−2.47	−4.37	−2.65
T-5	−67.86	−56.97	−14.96	−0.37	−6.57	−4.61	−6.74
T-6	−60.51	−47.66	−16.80	−0.31	−4.16	−4.48	−4.38
T-7	−51.05	−42.19	−12.26	−0.39	−3.74	−3.74	−3.69
T-8	−45.96	−35.30	−13.97	−0.33	−3.46	−3.69	−3.51
T-9	−66.07	−57.14	−12.37	−0.35	−3.69	−3.85	−3.75
T-10	−58.08	−47.35	−14.47	−0.31	−2.34	−4.18	−2.48
T-11	−63.42	−54.50	−12.61	−0.35	−4.10	−4.09	−4.15
T-12	−56.40	−45.65	−14.43	−0.29	−2.15	−4.14	−2.33
T-13	−47.07	−40.84	−8.40	−0.23	−2.39	−2.27	−2.26
T-14	−41.32	−34.00	−9.40	−0.19	−2.16	−2.17	−2.06
T-15	−62.12	−55.84	−8.47	−0.22	−2.39	−2.34	−2.32
T-16	−54.32	−47.02	−9.23	−0.17	−2.01	−2.00	−1.91
T-17	−59.29	−53.03	−8.63	−0.21	−2.63	−2.50	−2.55
T-18	−53.44	−46.08	−9.79	−0.17	−2.58	−2.56	−2.54
T-19	−46.64	−40.78	−7.98	−0.21	−2.33	−2.24	−2.24
T-20	−40.89	−33.97	−8.97	−0.18	−2.13	−2.15	−2.05
T-21	−61.71	−55.80	−8.05	−0.21	−2.35	−2.31	−2.31
T-22	−53.91	−47.01	−8.80	−0.17	−2.00	−1.98	−1.91
T-23	−58.84	−52.96	−8.18	−0.19	−2.56	−2.44	−2.51
T-24	−53.01	−46.06	−9.35	−0.15	−2.56	−2.53	−2.54
T-25	−43.72	−39.89	−5.14	−0.09	−1.44	−1.32	−1.36
T-26	−37.61	−33.13	−5.74	−0.07	−1.29	−1.26	−1.22
T-27	−58.74	−54.86	−5.14	−0.09	−1.41	−1.32	−1.38
T-28	−50.72	−46.21	−5.59	−0.06	−1.20	−1.11	−1.17
T-29	−55.96	−54.14	−5.37	−0.11	−3.74	−1.55	−1.63
T-30	−49.54	−45.06	−6.00	−0.09	−1.56	−1.52	−1.47
T-31	−43.48	−39.86	−4.90	−0.08	−1.41	−1.28	−1.33
T-32	−37.36	−33.11	−5.50	−0.05	−1.27	−1.24	−1.21
T-33	−58.51	−54.84	−4.91	−0.08	−1.39	−1.29	−1.36
T-34	−50.50	−46.21	−5.36	−0.06	−1.20	−1.10	−1.17
T-35	−55.60	−51.99	−5.05	−0.10	−1.59	−1.43	−1.48
T-36	−49.17	−43.62	−5.37	−0.06	−0.12	−1.11	−1.35

## Data Availability

Not applicable.

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
