# Peer review of "Chalcogen Bond Involving Zinc(II)/Cadmium(II) Carbonate and Its Enhancement by Spodium Bond"

_molecules, 2021, doi:10.3390/molecules26216443_

Round 1

Reviewer 1 Report

Dear Authors and Editors,

In general, the part concerning chalcogen bonding characterization is well-written and is scientifically correct. However, I have several serious questions:

1/ What is the general motivation of the study? In my opinion, the reported systems are very exotic as they do not match either the classical works devoted to the studies of group-named interaction in the gaseous phase or crystal engineering approaches dealing with the analysis of short contacts in the crystal structures. I hardly can imagine, how such complexes can ever take place, as carbonates are ionic hardly soluble crystals and can not form either co-crystals or gaseous complexes. 

2/ Why do the authors describe the discovery and properties of selenium as an element in the introduction section? How is it relevant to the topic of the article?

3/ The part characterizing the spodium bond is extremely problematic. In general, this bond is analyzed in case of metal-organic compounds or when the metal atom is connected to an electron-acceptor substituent to make a sigma-hole more pronounced. However, here we do not have strong covalent bonding and substituents promoting significant sigma-hole of the central metal. Moreover, the authors do not demonstrate the existence of the sigma-hole in the source carbonate at all. Besides in fig. 5, the authors provide different mutual orientations of metal and nitrogen-containing species and in all the cases name the interaction as spodium bond. 

In my opinion, this part of the work should be significantly reworked as well as the general motivation for the study of such exotic complexes.

Author Response

Comment 1: What is the general motivation of the study? In my opinion, the reported systems are very exotic as they do not match either the classical works devoted to the studies of group-named interaction in the gaseous phase or crystal engineering approaches dealing with the analysis of short contacts in the crystal structures. I hardly can imagine, how such complexes can ever take place, as carbonates are ionic hardly soluble crystals and can not form either co-crystals or gaseous complexes.
Response: Yes, carbonates are ionic hardly soluble crystals and can not form either co-crystals or gaseous complexes. Owing to their good properties and extensive applications in many fields [5-10], thus based on this model molecules, we focused on their interactions with other molecules by means of theoretical methods although the above weakness. Our motivation is to explore the chalcogen bond formed by MCO3 (M = Zn, Cd) and SeHX (X=F, Cl, OH, OCH3, NH2, NHCH3) to unveil the substituent effect on the strength of chalcogen bond and its origin as well as the enhancement of chalcogen bond by means of cooperativity with the spodium bond.

Comment 2: Why do the authors describe the discovery and properties of selenium as an element in the introduction section? How is it relevant to the topic of the article?

Response: The related text has been moved: Selenium (Se) element was discovered by Berzelius [15], and the first organic Se compound was prepared by Löwig [16]. Because of its soft metal-like behavior, Se plays an important role in biological processes, ligand chemistry, and asymmetric synthesis [17-19]. For instance, Se is a trace element necessary for the metabolism of organisms in which it plays a role in the immune system through selenoproteins [20]. In addition, Se has a unique redox activity, exhibiting significantly biochemical and pharmaceutical activities, such as anti-tumor, antibacterial, and anti-aging [21].

Comment 3: The part characterizing the spodium bond is extremely problematic. In general, this bond is analyzed in case of metal-organic compounds or when the metal atom is connected to an electron-acceptor substituent to make a sigma-hole more pronounced. However, here we do not have strong covalent bonding and substituents promoting significant sigma-hole of the central metal. Moreover, the authors do not demonstrate the existence of the sigma-hole in the source carbonate at all. Besides in fig. 5, the authors provide different mutual orientations of metal and nitrogen-containing species and in all the cases name the interaction as spodium bond.

Response: Yes, spodium bond is usually analyzed in case of metal-organic compounds. However, inorganic compounds such as MCl3- (M=Zn, Cd, Hg) [ChemPhysChem 2020, 21, 1119-1125], ZnCl2 [Molecules 2021, 26, 2498] are also used in studying the spodium bonds. In MCO3, no strong covalent bonding is present, but the positive MEPs are present on the M atomic surface (Figure S5). In this study, our aim is to study the chalcogen bond formed by MCO3 and its enhancement by the spodium bond. The spodium bond is not analyzed in this study since it has been studied [ChemPhysChem 2021, 22, 1698-1705]. The following sentences have been added: The formation of SpB can be understood by the MEP maps of MCO3 and three N-bases (Figure S5), where a red region with positive MEPs and a blue one with negative MEPs are found on the M and N atoms, respectively. Since such M···N SpB has been analyzed in the previous study [14], thus it is not studied here and our aim is to strengthen the chalcogen bond by means of SpB.

Reviewer 2 Report

This paper explores the interactions between XSeH and MCO3 [where M is either Zn(II) or Cd(II)] and how the chalcogen bond interaction is enhanced by interactions of M with nitrogen bases.  This paper is of interest and clearly written.  I have some comments for the authors to address before publication.

1)  The authors discuss chalcogen bonding with no reference to σ-hole bonding.  The authors should for completeness mention σ-hole bonding.  See Clark et al, J. Mol. Model. 13, 291 (2007), where the term "σ-hole" was introduced in relation to halogen bonding, and then see Murray et al, J. Mol. Model. 13, 1033 (2007), where it was extended to Group 16-containing molecules, the chalcogens.  See also PCCP, 15, 11178 (2013); 19, 3216 (2017), and references therein.

2)  It is not inherently surprising that the "spodium" bonds are stronger than the chalcogen bonds.  The Zn(II) and Cd(II) are metal cations in the neutral carbonate salts.

3)  The interactions discussed in this paper can be explained without invoking orbital considerations.  See Politzer et al, J. Mol. Model. 21, 52 (2015) and Clark et al, PCCP 20, 30076 (2018).  What is defined as charge transfer is an extreme form of polarization.

4)  There is interesting data in this paper.  I recommend that the authors focus on the physical observables.

Author Response

This paper explores the interactions between XSeH and MCO3 [where M is either Zn(II) or Cd(II)] and how the chalcogen bond interaction is enhanced by interactions of M with nitrogen bases. This paper is of interest and clearly written. I have some comments for the authors to address before publication.

Comment 1: The authors discuss chalcogen bonding with no reference to σ-hole bonding. The authors should for completeness mention σ-hole bonding. See Clark et al, J. Mol. Model. 13, 291 (2007), where the term "σ-hole" was introduced in relation to halogen bonding, and then see Murray et al, J. Mol. Model. 13, 1033 (2007), where it was extended to Group 16-containing molecules, the chalcogens. See also PCCP, 15, 11178 (2013); 19, 32166 (2017), and references therein.

Response: These references have been cited and discussed: The σ-hole bonding interaction was first utilized to describe halogen bonding [16]. Then such σ-hole bonding was extended to molecules containing group VI atoms [17]. This concept was further reviewed and revisited [18,19].

[16] Clark, T.; Hennemann, M.; Murray, J.S.; Politzer, P. Halogen bonding: The σ-hole. J. Mol. Model. 2007, 13, 291-296.

[17 Murray, J.S.; Lane, P.; Clark, T.; Politzer, P. σ-Hole bonding: Molecules containing group VI atoms. J. Mol. Model. 2007, 13, 1033-1038.

[18] Halogen bonding and other σ-hole interactions: A perspective. Phys. Chem. Chem. Phys. 2013, 15, 11178-11189.

[19] Politzer, P.; Murray, J.S.; Clark, T.; Resnati, G.; The σ-hole revisited. Phys. Chem. Chem. Phys. 2017, 19, 32166-32178.

Comment 2: It is not inherently surprising that the "spodium" bonds are stronger than the chalcogen bonds. The Zn(II) and Cd(II) are metal cations in the neutral carbonate salts.

Response: Yes, the "spodium" bonds are stronger than the chalcogen bonds since the Zn(II) and Cd(II) are metal cations in the neutral carbonate salts. This has been pointed out in the revised manuscript: The spodium bond is stronger than the chalcogen bond since the Zn(II) and Cd(II) are metal cations in the neutral carbonate salts.

Comment 3: The interactions discussed in this paper can be explained without invoking orbital considerations. See Politzer et al, J. Mol. Model. 21, 52 (2015) and Clark et al, PCCP 20, 30076 (2018). What is defined as charge transfer is an extreme form of polarization.

Response: The following discussion has been added and both references have been cited.

Politzer and co-authors explained most σ-hole interactions by means of Coulombic interactions and concluded that charge transfer is an extreme form of polarization [45,46]. However, the interaction energy of chalcogen bond displays a good linear relationship with the charge transfer but not with polarization energy.

[45] Clark, T.; Murray, J.S.; Politzer, P. A perspective on quantum mechanics and chemical concepts in describing noncovalent interactions. Phys. Chem. Chem. Phys. 2018, 20, 30076-30082.

[46] Politzer, P.; Murray, J.S.; Clark, T. Mathematical modeling and physical reality in noncovalent interactions. J. Mol. Model. 2015, 21, 52.

Comment 4: There is interesting data in this paper. I recommend that the authors focus on the physical observables.

Response: The strength of chalcogen bond in the binary systems and the enhancing mechanism of chalcogen bond by the spodium bond in the ternary systems have been explained with electrostatic potentials, which are physical observables.